# Photocatalytic Removal of Antibiotics from Wastewater Using the CeO_2_/ZnO Heterojunction

**DOI:** 10.3390/ma16020850

**Published:** 2023-01-15

**Authors:** Nicolae Apostolescu, Ramona Elena Tataru Farmus, Maria Harja, Mihaela Aurelia Vizitiu, Corina Cernatescu, Claudia Cobzaru, Gabriela Antoaneta Apostolescu

**Affiliations:** Department of Chemical Engineering, “Cristofor Simionescu” Faculty of Chemical Engineering and Environmental Protection, “Gheorghe Asachi” Technical University of Iasi, 73, Prof.dr.doc. Dimitrie Mangeron Avenue, 700050 Iasi, Romania

**Keywords:** photocatalyst, CeO_2_/ZnO heterojunction, chlortetracycline, ceftriaxone

## Abstract

CeO_2_/ZnO-based photocatalytic materials were synthesized by the sol-gel method in order to establish heterojunctions that increase the degradation efficiency of some types of antibiotics by preventing the recombination of electron–hole pairs. The synthesized materials were analysed by XRD, SEM, EDAX, FTIR, and UV-Vis. After several tests, the optimal concentration of the catalyst was determined to be 0.05 g‧L^−1^ and 0.025 g‧L^−1^ for chlortetracycline and 0.05 g‧L^−1^ for ceftriaxone. CeO_2_/ZnO assemblies showed much better degradation efficiency compared to ZnO or CeO_2_ tested individually. Sample S3 shows good photocatalytic properties for the elimination of ceftriaxone and tetracycline both from single solutions and from the binary solution. This work provides a different perspective to identify other powerful and inexpensive photocatalysts for wastewater treatment.

## 1. Introduction

In our constantly and rapidly evolving world, finding solutions to protect the environment has been a priority lately. Among the substances that improve our life, but also make it more difficult, antibiotics are indispensable products due to the positive health benefits, though they have negative effects on natural ecosystems. Antibiotics are necessary medicines; many of them are on the WHO list of essential medicines, and their consumption is continuously increasing [1,2]. Unfortunately, some of these substances are excreted from the body unmetabolized in percentages that vary depending on the compound (up to 80% for tetracycline [3]), and their accumulation in wastewater and soil causes complications [4,5,6,7,8]. The presence of antibiotics in wastewater and also in drinking water has been detected in small amounts, but this is quite alarming for environmental safety [2,9]. The use of antibiotics in excess also leads to the appearance of antibiotic-resistant microorganisms [5,6,10,11]. Antibiotic resistance is a current problem leading to a decrease in the effectiveness of these classes of drugs. Tetracycline and ceftriaxone are widely used to treat various bacterial infections and have been detected in wastewater at varying concentrations. They are stable over time in the complex wastewater mixtures and are difficult to remove using classic water treatment methods; they can adversely affect the wastewater treatment processes (for example the ammonia removal process or chemical oxygen demand removal). There are several ways to remove antibiotics and other contaminants from wastewater, such as adsorption, sonochemical processes, ozonation, membrane technology, aerobic or anaerobic treatment, phytoremediation, biogeochemical methods such as wetlands, chemical disinfection methods such as chlorination, electrochemical oxidation processes, osmosis and electro-osmosis, electro-flocculation, ionizing radiation, and hybrid technologies [8,10,12,13,14,15,16,17,18,19,20,21,22,23,24]. However, it seems that the most effective methods of reducing the level of antibiotics in water are the methods that involve photocatalysis and advanced oxidation processes. Photocatalytic methods are preferable to others because they are financially inexpensive and effective at low antibiotic concentrations, and the photocatalytic materials used are stable and environmentally benign [25,26,27,28,29,30,31]. Over time, several semiconducting materials have been tested for photocatalytic activity, but good results have been obtained showing oxides, sulphides, and their composites. Oxide materials were chosen because they are stable and inert, and the combination of two oxides leads to improved semiconductor properties. Although the most studied photocatalytic material is TiO_2_, the results obtained with other materials have shown promising results. Photocatalytic systems based on oxides from blocks d and f (with semiconductor properties) have given good results in eliminating or reducing some polluting compounds from wastewater. Semiconductor materials are particularly attractive for environmental remediation applications. If semiconductors are exposed to electromagnetic radiation with an energy intensity greater than or close to the value of the band gap (E_g_), electron–hole pairs are generated (valence band–conduction band); as a consequence (e^−^–h^+^) pairs are able to initiate redox reactions. According to studies in the literature, a photocatalyst’s activity can be increased by doping, which is the addition of a substance that modifies the band gap value slightly in regions where incident radiation (ultraviolet or visible) can be used efficiently, by forming heterostructures, or by including reaction accelerators in the system that have the purpose of producing more free radicals or preventing their recombination [31,32,33]. The photocatalytic activity is closely related to the physicochemical properties but also to the morphology and texture of the materials studied, so the synthesis techniques are often of great importance. In this study, we followed the efficiency of some zinc oxide and cerium dioxide heterostructured materials synthesized in our laboratory towards the photocatalytic degradation of a classical antibiotic (chlortetracycline) and a third-generation cephalosporin-type antibiotic—ceftriaxone. Cerium has exciting catalytic characteristics because 4d and 5p electrons sufficiently defend the 4f orbitals. Ceria or Cerium(IV) oxide is a versatile, inert, and physically and chemically stable material with multiple and diverse applications. Due to its hardness (Mohs scale 7), it was initially used as an abrasive material, but today it is used (alone or in binary or complex mixtures) in the field of heterogeneous catalysis (oxidation of hydrocarbons) or in the field of sensors, energy, and fuels such as solid oxide fuel cells, but also in water-splitting processes or photocatalysis [33,34,35]. CeO_2_ applications in the dermato-cosmetics industry and in the biomedical field (antibacterial effect) should also be mentioned here [36,37,38]. It is also possible to combine two or more properties, such as the infrared filtering properties with the photocatalytic ones, to optimize practical applications. Zinc oxide also has physical and chemical properties that ensure numerous practical uses, since it is introduced into the composition of many products to improve their properties (e.g., UV or antimicrobial protection); ZnO (alone or together with other oxides) is used in environmental applications, in medical applications, or as a sensor [32,39,40,41]. Using ZnO or CeO_2_ separately leads to fast recombination processes that decrease the photocatalytic efficiency. The reduction in recombination rate can occur through the combination of two or more oxides as new energy bands are formed. Zinc oxide and cerium dioxide work well together and, in most cases, have a synergistic effect when they are associated in practical applications. [34,40,41].

As we know, there are several available studies on CeO_2_/ZnO applications, as heterostructures as well as individually [20,24,32,38]. However, here, PVA, Poly(vinyl alcohol) was used as a dispersant for heterostructure management, and the obtained material showed good results for the photocatalytic degradation of two antibiotics, separately and in admixture, for low doses of UV radiation and for a small amount of catalyst. The results of this study can serve as a starting point for further research on CeO_2_/ZnO materials.

## 2. Materials and Methods

### 2.1. Reagents and Preparation

Photocatalytic materials were obtained by hydrothermal synthesis [42,43] starting from Zn(CH_3_COO)_2_·2H_2_O and Ce(NO₃)₃‧6H₂O (Sigma-Aldrich) as precursors and NH_4_OH (25% from Chemical Company, Romania) and PVA (average Mw 89,000–98,000, 99%, Aldrich) as anti-agglomeration agents. The precursors were mixed with ammonia solution in different molar ratios (S2- 2ZnO:CeO_2_; S3- ZnO:2CeO_2_), the mixture was kept at 120 °C for 8 h in a Teflon autoclave, and then the precipitates obtained were separated by centrifugation and washed with bidistilled water until the total elimination of NH_4_OH, NH_4_NO_3_, and CH_3_COOH. The precipitates thus obtained (cerium and zinc hydroxide) were immersed in a 0.001 mol‧L^−1^ PVA solution for 30 min, filtered, and then calcined at 400 °C (at a heating rate of 5 grd‧min^−1^). Polyvinyl alcohol has the task of avoiding the agglomeration of particles and is eliminated by calcination. The samples obtained were named S1: ZnO, S2: 2ZnO-CeO_2_, S3: ZnO-2CeO_2_, and S4: CeO_2_.

The antibiotics tested were chlortetracycline and ceftriaxone (Sigma Aldrich), presented in Table 1.

### 2.2. Sample Characterization

The synthesized samples were characterized by X-ray diffraction (Rigaku, Tokyo, Japan) and CuKα radiation (2θ angle, range from 10 to 80; step 0.02°/s); the possible functional groups remaining on the surface of the photocatalyst were identified by FTIR (Perkin Elmer Spectrum 100, PerkinElmer Inc., Shelton, CT, USA), resolution 2 cm^−1^ using 32 scans in the range 4000–400 cm^−1^; all samples were prepared as KBr pellets (ratio 5/95 wt.%). The morphology of the prepared samples was observed with a Quanta 200 scanning electron microscope equipped with an energy-dispersive X-ray spectroscopy analyzer (Bruker Optics Inc, Billerica, MA, USA). The UV-Vis absorption spectra of the solid samples were obtained with a Jasco V-550 device (Jasco International CO, Kyoto, Japan) equipped with an integrating sphere. Monitoring of the photocatalytic degradation of antibiotics was performed also with a Jasco V-550 device (Kyoto, Japan). 

### 2.3. Photodegradation Experiments

The magnetically stirred aqueous suspensions were UV-irradiated in a flat cylinder reactor (total volume: 100 cm^3^) exposed to air. The radiant flux entering the reactor was about 0.21 mW‧cm^−2^ (Hamamatsu C9536-01 m with H9958 detector for 310–380 nm), calculated from the distance between the samples and the light source produced by a UV-B lamp with Hg (18 W) (OSRAM, Munich, Germany). The volume of the solution was 75 cm^3^, and the catalyst dose was 0.05 g‧L^−1^. Aqueous solutions were prepared using deionized bidistilled water (Milli-Q, Millipore, Darmstadt, Germany). The degradation operations were carried out at room temperature at natural pH. The aerated suspension was first stirred in the dark for 40 min, which was sufficient to achieve equilibrated adsorption. The initial concentration was 0.025 mg‧mL^−1^ chlortetracycline and ceftriaxone 0.05 mg‧mL^−1^.

The tests were performed without changing the pH of the native antibiotic solution. Samples were taken at fixed timed intervals and centrifuged to remove the solid, then the absorbance of the supernatant was read three times, noting the mean value. The level of antibiotic degradation was quantified using the correlation Equation (1):(1)$$R\left(\%\right)=\frac{\left(A_{0}{\;-\;A}_{i}\right)}{A_{0}}\cdot 100$$
where R(%) is the antibiotic degradation yield, A_0_ and A_i_ are the initial and t_i_ time antibiotic absorbance values at the same time values.

## 3. Results and Discussion

### 3.1. Characterization of the Photocatalysts

The X-ray diffraction spectra of ZnO and CeO_2_ and those of the as-synthesized composites are shown in Figure 1. They present several peaks that could be indexed in accordance with the diffraction spectrum of ZnO (JCPDS card no. 36-1451) and CeO_2_ (JCPDS card no. 34-0394). For S1 (ZnO), the peaks were positioned at the 2θ angles: 31.83, 34.51, 36.32, 47.61, 55.64, 62.94, 68.03, and 69.16 can be indexed as (100), (002), (101), (102), (102), (103), (112), and (210), indicating a hexagonal wurtzite phase of ZnO; for S4 (CeO_2_), the peaks were positioned at the 2 angle: 28.55, 56.44, 59.22, 69.37, 76.79, 79.11, and 88.48, which can also be indexed as (111), (311), (222), (400), (311), (420), and (422). The CeO_2_ (111) peak is associated with the cubic structure. Samples S2 and S3 show the peaks corresponding to the polycrystalline structures of pure oxides, indicating the formation of composite materials with no other impurities [47,48,49,50].

In order to get more information about the appearance of the surfaces of the prepared samples, the SEM (Scanning Electron Microscopy) images were examined. The SEM electron microscopy images showing the morphology of the surfaces at different magnifications and the EDAX (Energy Dispersive X-ray) profiles are shown in Figure 2. Samples S1–S4 present porous structures with different surface morphologies, from small and uneven aggregates for S1–S3 to aggregates with an isometric structure for S4. The boundaries between the particles are not well defined.

For S4, crystallite size varies from below 0.1 µm × 0.1 μm × 0.1 μm to approximately 1 μm × 2 μm × 3 μm. The accumulation of micro-crystallites is a normal process; the crystallites try to reach the minimum energy state, minimizing the contact area with the external environment. The small size of the obtained crystallites explains the good photocatalytic activity. EDAX analysis confirms the existence of Zn, Ce, and O elements, so CeO_2_ and ZnO oxides are present. 

The FTIR spectra are presented in Figure 3. FTIR analysis confirms that the organic phase has been eliminated by calcination. The peaks in the 3400–3450 cm^−1^ range are due to adsorbed water molecules (the O-H bond stretching vibration), and those in the 550–400 cm^−1^ range are generated by the vibrations of the metal oxide bonds [49,51].

Next, solid-state UV-Vis spectroscopy was used to obtain information on the optical properties of the synthesized samples and to calculate the E_g_ values (Figure 4). The optical band gap was determined using the Tauc formula, ${\left(\mathrm{ah}\nu \right)}^{2}=A\left({h\nu \;-\;E}_{g}\right)$, in which a is the absorption coefficient, A is a constant, h is Planck’s constant, ν is the frequency of incident radiation, E_g_ is the optical band gap, and n = 1/2 (for ZnO and CeO_2_) [48,52]. The E_g_ values are obtained by extrapolating the straight lines to the point of intersection with the *x*-axis. Compared to pure ZnO and pure CeO_2_, the composite materials S2 and S3 have a different E_g_ value.

The observed decrease in E_g_ values can be explained by the occurrence of numerous surface defect states such as oxygen vacancies, the coexistence of Ce^4+^ and Ce^3+^ in the CeO_2_/ZnO heterostructure, and the interaction between ZnO and CeO_2_ nanocrystals [53,54].

### 3.2. Antibiotic Photocatalytic Degradation

To evaluate the photocatalytic activity, the synthesized samples were contacted with solutions of chlortetracycline, ceftriaxone, and a chlortetracycline–ceftriaxone mixture (Figure 5, Figure 6 and Figure 7). Work was carried out without adjustments to the natural pH value of the solutions. The doses of photocatalytic material have been studied previously; only the results for the 0.05 g‧L^−1^ concentration are presented here. Experiments performed with UV irradiation but without a photocatalyst showed that both antibiotics are relatively stable to UV exposure. The experimental results have similar profiles; only a small fraction of the antibiotics is degraded after exposure to ultraviolet light in the absence of photocatalysts. The substantial increase in the photocatalytic performance of samples S2 and S3 compared to S1 and S4 (pure oxides) is due to the process of delaying the recombination of electron–hole pairs, owing to the formation of heterojunctions between the two oxides; this is advantageous for keeping the promoted electron in the conduction band of ZnO for a longer period of time. In this situation, adsorbed oxygen is more likely to form superoxide O_2_^−^ radicals.

When CeO_2_/ZnO —based photocatalytic materials are irradiated with UV rays, pairs of electric charge holes in the valence band and electric charge electrons in the conduction band are formed. The holes immediately react with water molecules or hydroxyl ions and form hydroxyl radicals, which are very strong oxidizing agents of organic molecules, according to the following Equations (2)–(9) [29,43,49]:(2)$$\mathrm{ZnO}\;+\;h\upsilon \;\to \;\mathrm{ZnO}\left(e_{\left(\mathrm{cb}\right)}^{-}{\;+\;h}_{\left(\mathrm{VB}\right)}^{\;+\;}\right)$$
(3)$${\mathrm{CeO}}_{2}\;+\;h\upsilon \;\to {\;\mathrm{CeO}}_{2}\left(e_{\left(\mathrm{cb}\right)}^{-}{\;+\;h}_{\left(\mathrm{VB}\right)}^{\;+\;}\right)$$
(4)ZnOe− + O2 → O2− + ZnO→   H2O O ·H + ZnO
(5)ZnOh+ + OH−→ O ·H + ZnO
(6)CeO2e− + O2 → O2− + CeO2→ H2O O ·H + CeO2
(7)CeO2h +  + OH− → O ·H + CeO2
(8)$${\mathrm{Ce}}^{4+}{\;+\;e}_{\left(\mathrm{CB}\right)}^{-}\;\to {\;\mathrm{Ce}}^{3+}$$
(9)O ·H + O2− + hVB +  + Antibiotic → degradation products

Oxidation processes of antibiotics take place in hole species (h^+^) and O_2_^−^. Photocatalytic activity is influenced by crystal structure, specific surface area, particle size distribution, porosity, surface hydroxyl group density, etc. All these properties affect the formation of electron–hole pairs, the adsorption–desorption surface area, and the redox process. The photocatalytic activity of the studied materials was enhanced by delaying the recombination of electron–hole pairs. The main method of slowing down is through the formation of heterojunctions in the CeO_2_/ZnO mixture (Figure 8), a process described in other studies [49,52,53].

The electrons in the 4f orbitals of Ce interfere with the 3d electrons of Zn and the 2p electrons of O, resulting in the formation of a new band between BV and BC that changes the characteristics of the oxide mixture. The highest level of degradation was achieved for CFTX at 71.23% in the presence of S3, followed by CT at 58.65%, also for S3 (Figure 9). The higher proportion of CeO_2_ in the synthesized materials has resulted in a higher degradation rate; Ce^4+^ ions act as a trap to prevent the recombination of electron–hole pairs generated by irradiation with ultraviolet rays, with E > E_g_; thereby, the photocatalytic process is accelerated. 

Compared to pure CeO_2_ and pure ZnO, the superior catalytic performance of samples S2 and S3 is attributed to the formation of heterojunctions, which are an effective method for modifying the properties of mixed oxides.

The information gathered in Table 2 reveals the large diversity of materials that can be used as photocatalytic materials for ceftriaxone and tetracycline degradation.

### 3.3. Kinetic Analysis

The photocatalytic process was assumed to follow a pseudo-first-order model according to Equation (10).
(10)$$-k_{\mathrm{obs}}\cdot t=\ln\left(C_{t}-{\;C}_{0}\right)=\ln\left(A_{t}-{\;A}_{0}\right)$$
where A_0_ and A_t_ represent the absorbance of antibiotics solution measured at specific λ, initially and at t moment, and k_obs_ is the first-order oxidation rate constant (min^−1^) [28,29,51]. The kinetic curves of the photocatalytic degradation of the two antibiotics with the prepared samples in 120 min are shown in Figure 10, Figure 11 and Figure 12. According to this equation, if the experimental data show first-order kinetics, a line in the coordinates −ln(A/A_0_) vs. t must be obtained. Therefore, a pseudo-first-order kinetic model was used to fit the experimental data presented in Figure 10, Figure 11 and Figure 12. Fitting of the rate data using a higher reaction order does not generate good coefficients.

## 4. Conclusions

In this work, we have reported a CeO_2_/ZnO mixed oxide powder with good photocatalytic activity for the degradation of some antibiotics (ceftriaxone and chlortetracycline) that are relatively stable to UV radiation. Four samples were synthesized by the hydrothermal method, two of which were pure oxides (ZnO and CeO_2_) and two of which were mixed oxides with different molar ratios (CeO_2_/ZnO). These were characterized by XRD, FTIR, SEM + EDAX, and UV-Vis on the solid. All four samples showed photocatalytic activity to reduce the level of antibiotics in wastewater (CFTX, CT, and the mixture of the two); the experimental results showed that mixed oxides behave better than pure oxides (Ce:Zn ratio = 2.1) and have better photocatalytic activity in all three tested situations (CFTX, CT, CFTX + CT). Since the radiation dose used is very low and the synthesized materials are chemically inert, we consider that they are useful for reducing the level of antibiotics in wastewater. The heterogeneous photocatalytic processes have a net advantage in the reduction in some contaminants from the aqueous medium (antibiotics), including the non-selective breakdown of pollutants to very low concentrations, normal pressure and temperature, use of oxygen as the primary oxidant, and the possibility of simultaneously inducing both oxidation reactions and reduction reactions.

## Figures and Tables

**Figure 1 materials-16-00850-f001:**
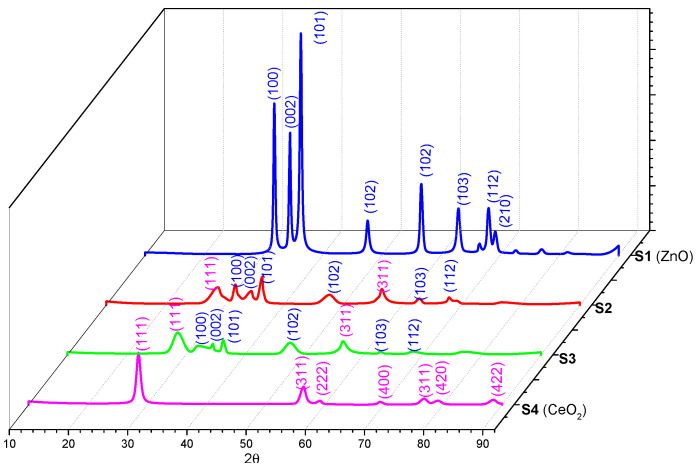
XRD patterns of the S1–S4 samples.

**Figure 2 materials-16-00850-f002:**
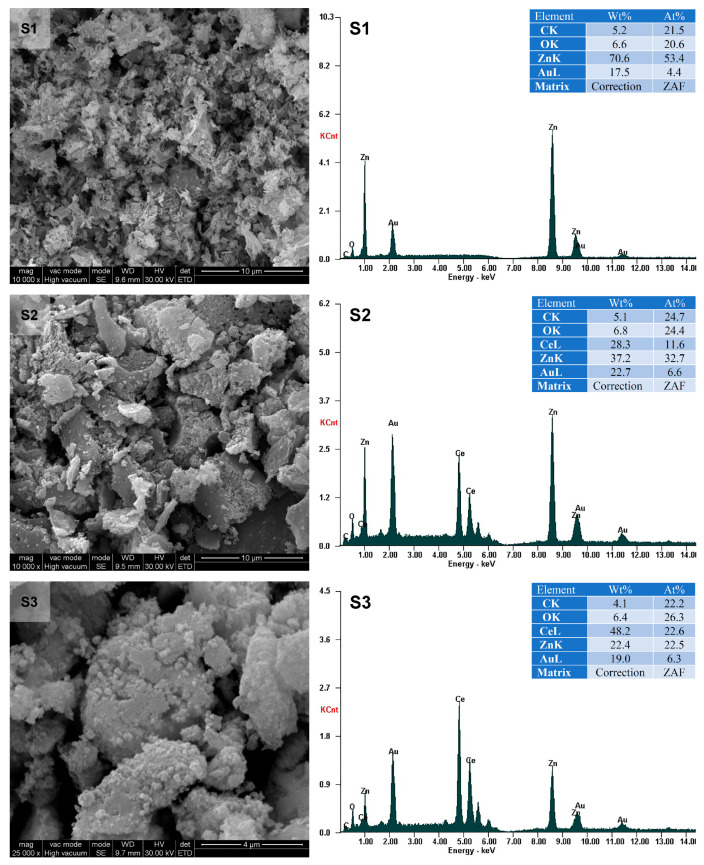
SEM micrographs (**left**) and EDAX pattern of the S1–S4 samples (**right**).

**Figure 3 materials-16-00850-f003:**
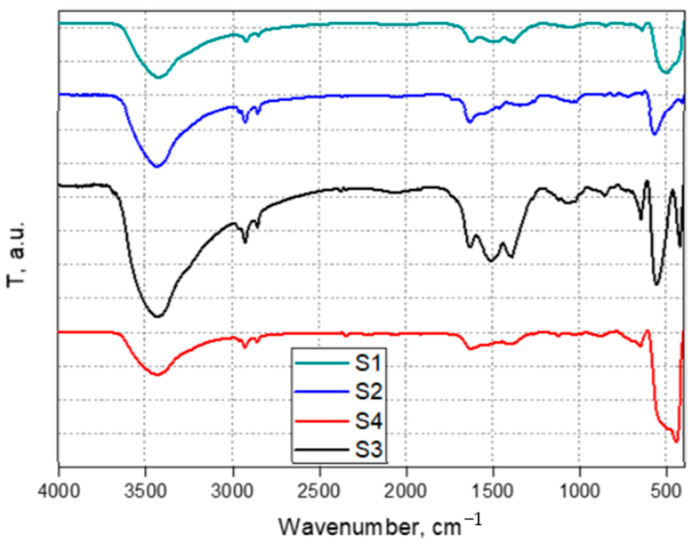
FTIR analysis of the synthesized samples.

**Figure 4 materials-16-00850-f004:**
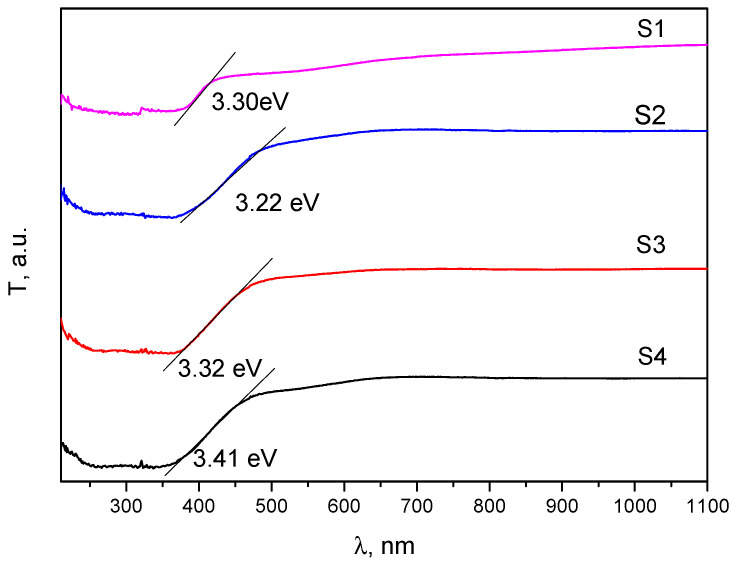
The optical absorption energy band gap estimated for CeO_2_/ZnO samples.

**Figure 5 materials-16-00850-f005:**
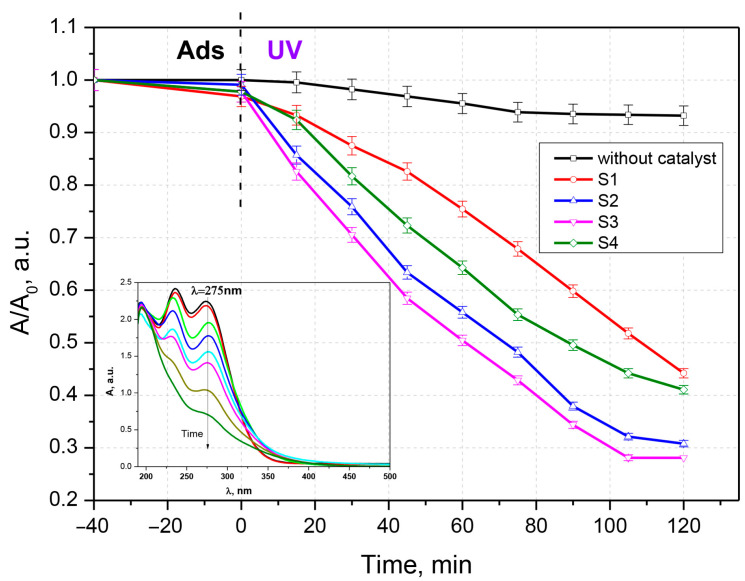
Photocatalytic degradation of chlortetracycline and time evolution of the UV-Vis absorption spectrum of chlortetracycline.

**Figure 6 materials-16-00850-f006:**
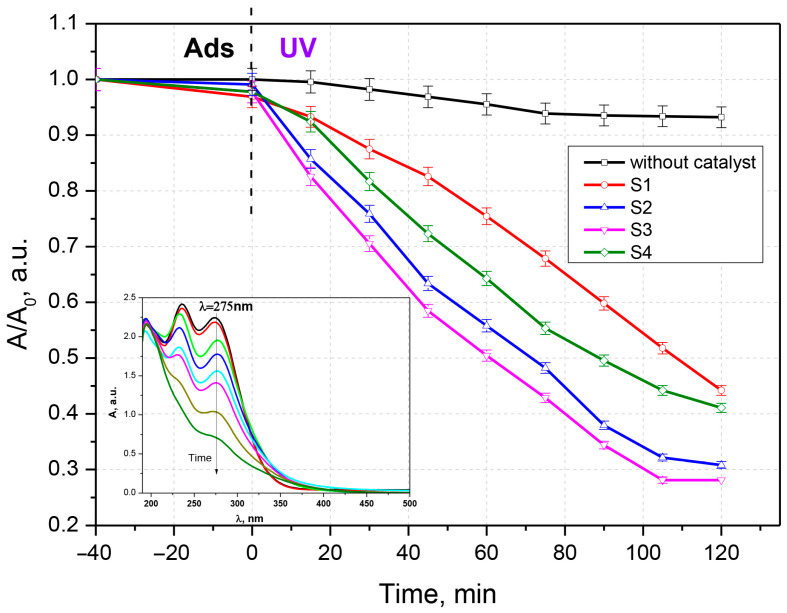
Photocatalytic degradation of ceftriaxone (**a**) and time evolution of the UV-Vis absorption spectrum of ceftriaxone.

**Figure 7 materials-16-00850-f007:**
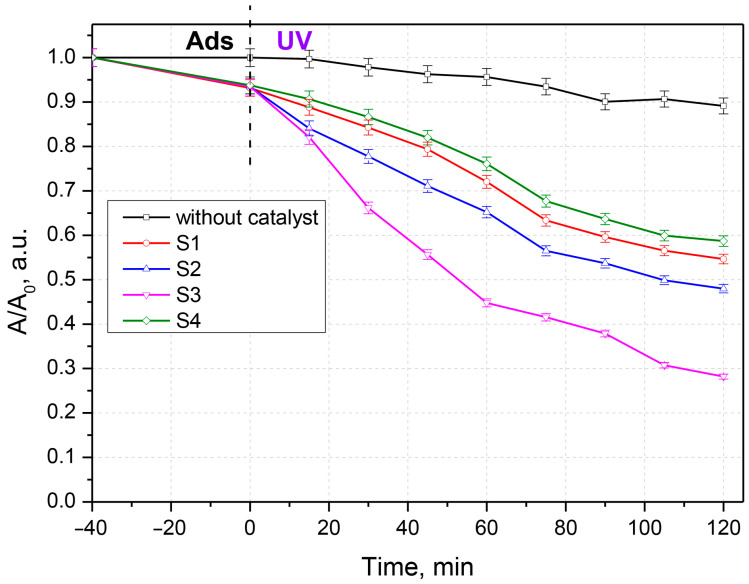
Photocatalytic degradation of the chlortetracycline + ceftriaxone mixture.

**Figure 8 materials-16-00850-f008:**
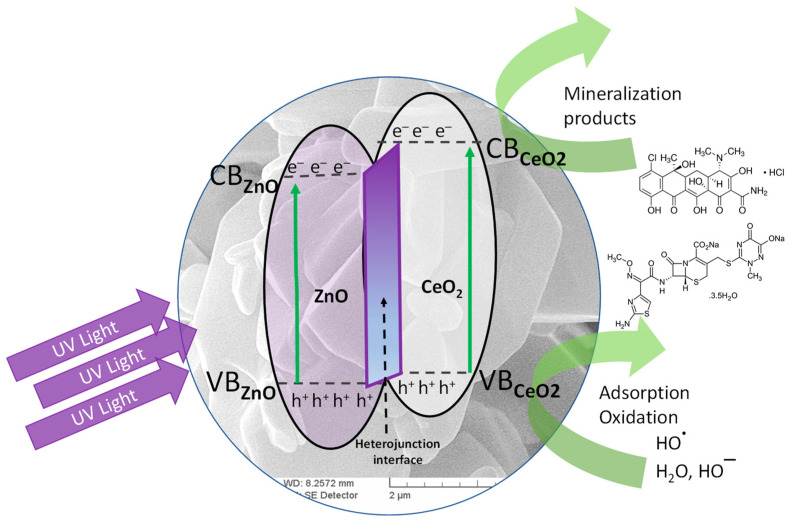
Proposed reaction mechanism for the photocatalytic activity of CeO_2_/ZnO heterojunctions.

**Figure 9 materials-16-00850-f009:**
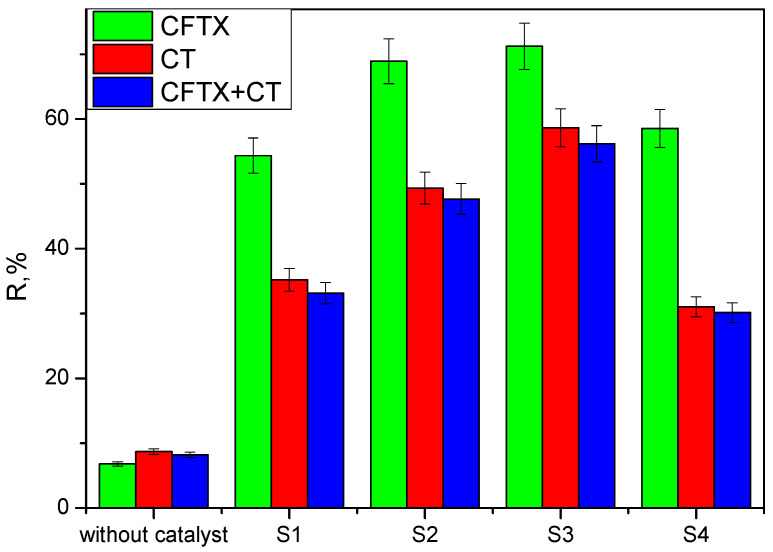
Degradation level for CFTX (ceftriaxone), CT (chlortetracycline) CFTX + CT (ceftriaxone+ chlortetracycline).

**Figure 10 materials-16-00850-f010:**
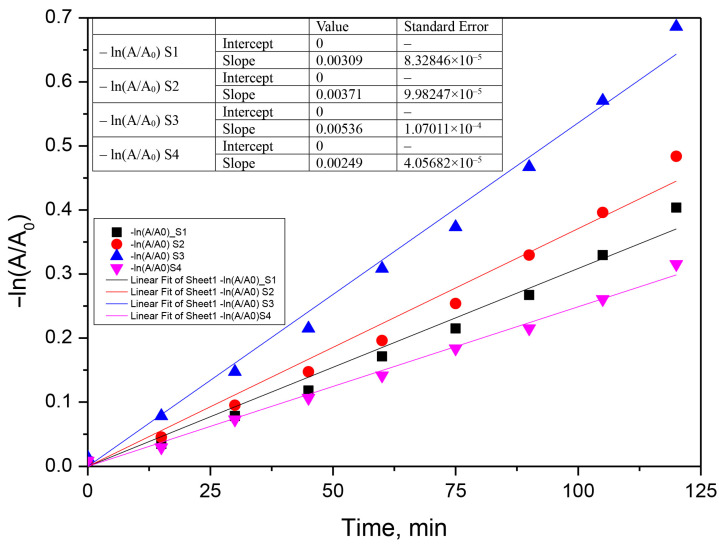
Fits of the pseudo-first-order kinetic model for the photocatalytic chlortetracycline degradation using the S1–S4 photocatalyst (0.05 g‧L^−1^, chlortetracycline concentration 0.025 g‧L^−1^).

**Figure 11 materials-16-00850-f011:**
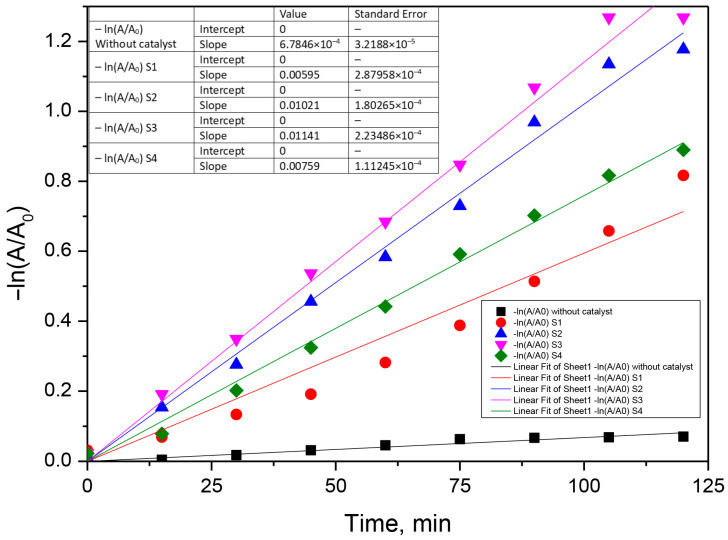
Fits of the pseudo-first-order kinetic model for the photocatalytic ceftriaxone degradation using the S1–S4 photocatalyst (0.05 g‧L^−1^, ceftriaxone concentration 0.05 g‧L^−1^).

**Figure 12 materials-16-00850-f012:**
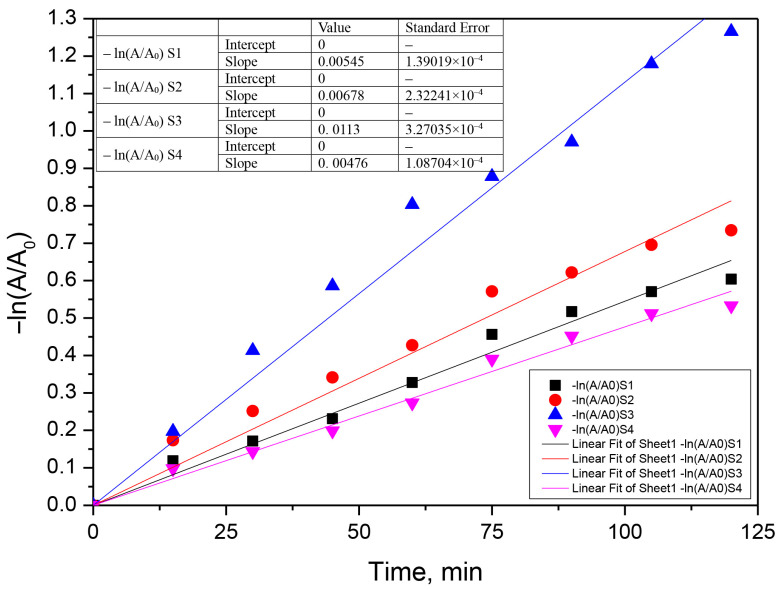
Fits of the pseudo-first-order kinetic model for the photocatalytic chlortetracycline + ceftriaxone mixture degradation using the S1–S4 photocatalyst (0.05 g‧L^−1^, chlortetracycline concentration 0.025 g‧L^−1^, ceftriaxone concentration 0.05 g‧L^−1^).

**Table 1 materials-16-00850-t001:** Chemical structure and some characteristics of antibiotics.

Chemical Composition and Structure	Properties
CT—Chlortetracycline hydrochloride C_22_H_23_ClN_2_O_8_‧HCl 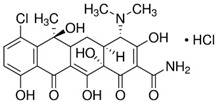	The first antibiotic discovered and used; 515.34 g‧mol^−1^ Solubility: 0.5–0.6 mg‧mL^−1^ (20 °C) Biological half-life: 5.6–9 h Excretion mode: 60% renal, less than 10% biliary [44] Toxicology: 2.31 mg‧kg^−1^ (LD50, mouse, oral) [45]
CFTX—Ceftriaxone C_18_H_16_N_8_Na_2_O_7_S_3_·3.5H_2_O 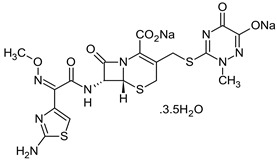	A third-generation antibiotic of the cephalosporin family; 661.60 g‧mol^−1^ Melting point: 155 °C Solubility: 0.105 g·L^−1^ Biological half-life: 5.8–8.7 h Excretion mode: 33–67% renal, 35–45% biliary [46]

**Table 2 materials-16-00850-t002:** Summaries of the literature on the photodegradation of ceftriaxone and tetracycline.

Antibiotic (Target Pollutant)	Catalyst Type	Irradiation Source Power Intensity, Exposure Time, Mineralization Degree	Ref
Ceftriaxone	ZnO nanospherical particles supported TiO_2_-nanorod MXene	Solar simulator 2000 550 W Max Lamp, 100 mW‧cm^−2^, 99.4%	[47]
Ceftriaxone C_0_ = 16.5–66 μM	Bi_2_WO_6_ and g-C_3_N_4_ nanosheets	KrCl excilamp, 222 nm 23 W, incident irradiance—0.74 mW‧cm^−2^, 60 min	[55]
Ceftriaxone 20–100 mg∙L^−1^	Fenton-like oxidation process, persulphate activator, iron dosage—0.1–0.5 g‧L^−1^ + scavengers (tert-butyl alcohol and isopropanol)	60 min, pH influence: 54.4% (pH: 5–6, Fe^2+^ dosage: 0.2 g‧L^−1^, PS concentration: 3 mM, initial antibiotic concentration: 20 mg‧L^−1^, UV power: 8 W: 20 °C) 95.7% (pH: 4.0, Fe^2+^ dosage: 0.3 g‧L^−1^, PS concentration: 4 Mm, UV, power: 8 W, 20 °C)	[27]
Ceftriaxone	Bi_2_WO_6_/g-C_3_N_4_	300 W Xe lamp, 120 min, 94.5%	[29]
Tetracycline, Oxytetracycline, Chlortetracycline	MoSSe nanohybrids	60 min 48.6% for TC 51.1% for OTC 56.5% for CTC	[56]
Oxytetracycline	MgAl calcined hydrotalcites	Pen Ray Power Supply 2.16 W MgAl-2.0, 59.32% for 5 h MgAl-2.5, 65.82% for 5 h MgAl-3, 63.87% for 5 h	[57]
Tetracycline	CeO_2_-ZnO hetero photocatalyst	300 W Xenon lamp, 60 min, 87.25%	[53]
Tetracycline	La_2_Ti_2_O_7_/AK—acid-modified coal-bearing strata kaolinite	300 W Xenon lamp La_2_Ti_2_O_7_, 60 min, 57.11% La_2_Ti_2_O_7_/CK, 60 min, 83.07% La_2_Ti_2_O_7_/AK, 60 min, 88.61%	[58]

## Data Availability

The data presented in this study are available on request from the corresponding authors.

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
