# Peer review of "Photocatalytic Removal of Antibiotics from Wastewater Using the CeO2/ZnO Heterojunction"

_materials, 2023, doi:10.3390/ma16020850_

Round 1
Reviewer 1 Report
Еhere are currently a lot of published studies on the use of CeO2/ZnO heterostructures for the photocatalytic removal of various contaminants, including antibiotics. I think that the novelty of the research is not sufficiently provided in the article. Authors should seriously explain how their study differs from existing ones. Although the studies presented in this manuscript have been carefully conducted and structured correctly, I cannot recommend this manuscript for publication. The idea of obtaining CeO2/ZnO heterostructures and using them to oxidize antibiotics is not new. In addition, the introduction does not present the problems of obtaining and activity of such heterostructures. There is also no comparison with similar articles. It would be better to write CeO2/ZnO heterostructures in the title of the article. Also, in the EDAX tables, the elements are given incorrectly.
Author Response
Response to comments of Reviewer 1 and list of changes made in the manuscript:
"Photocatalytic removal of antibiotics from wastewater using the CeO2/ZnO heterocatalyzers"
"Photocatalytic removal of antibiotics from wastewater using the CeO2/ZnO heterostructures"
We would like to thank the reviewer for comments and suggestions for improving the scientific quality of our manuscript. We have carefully considered all the reviewer comments and have revised the manuscript in light of them. The suggested modifications were clearly marked in red in the revised manuscript. Details of our responses to reviewer comments are shown below. We hope you will find these revisions rise to your expectations
Reviewer 1
There are currently a lot of published studies on the use of CeO2/ZnO heterostructures for the photocatalytic removal of various contaminants, including antibiotics. I think that the novelty of the research is not sufficiently provided in the article. Authors should seriously explain how their study differs from existing ones. Although the studies presented in this manuscript have been carefully conducted and structured correctly, I cannot recommend this manuscript for publication
The novelty of the research lies in the use of PVA as a structure directing agent, the low dose of radiation used (see inset table) and the simultaneous photodegradation of the two antibiotics.
The idea of obtaining CeO2/ZnO heterostructures and using them to oxidize antibiotics is not new. In addition, the introduction does not present the problems of obtaining and activity of such heterostructures. There is also no comparison with similar articles. It would be better to write CeO2/ZnO heterostructures in the title of the article.
We thank the Reviewer 1 for this useful remark. This correction was done in the revised manuscript. we inserted "heterojunction" in the title of the article
Also, we have included Table 2 in the text comparing the photodegradation parameters of ceftriaxone and tetracycline on various other semiconductor materials (table 2, line 254) and the relevant bibliographies (lines 413-429).
Also, in the EDAX tables, the elements are given incorrectly.
We thank again the Reviewer 1 for this important and useful remark. Accordingly Figure 2 was completed with the missing information At the suggestion of Reviewer 2, I changed the number of decimal places in the EDS images in Figure 2 (1 decimal place). During processing the samples were covered with colloidal gold and the corresponding signal appears in the EDX spectrum.

Reviewer 2 Report
The manuscript "Photocatalytic removal of antibiotics from wastewater using the CeO2/ZnO heterocatalyzers" represents some interesting aspects of highly functional hybrid photocatalyst heterostructures. In this work, CeO2/ZnO-based photocatalytic materials were synthesized by the sol-gel method in order to establish heterojunctions that increase the degradation efficiency of some types of antibiotics by preventing the recombination of electron-hole pairs. The synthesized materials were analyzed by XRD, SEM, EDAX FTIR and UV-VIS. Modification of bandgap structure is investigated and correlated to the catalysts properties. The authors have very perfectly represented the work with interesting data and discussions. The analyses are very interesting for the readers in this field of stuyd. I strongly recommend this manuscript for publication. The manuscript, however, requires some mandatory revisions before being considered for publication.
1. The introduction on similar works and more specifically on the implications of CeO2 and the implications of heterostructures for the catalytic activities needs a revision. I strongly recommend authors consult following manuscripts in their revision:
- Total Oxidation of Light Alkane over Phosphate-Modified Pt/CeO2 Catalysts. Environmental Science & Technology, 56 (13), 9661-9671, 2022,
doi: 10.1021/acs.est.2c00135
- Non-free Fe dominated PMS activation for enhancing electro-Fenton efficiency in neutral wastewater. Journal of Electroanalytical Chemistry, 928, 117062, 2023
doi: https://doi.org/10.1016/j.jelechem.2022.117062
2. I wonder if you have any data on the photocatalytic activity of your samples under visible light radiation.
3. In Figure 2, please report EDS data in one digit accuracy (5.2 for instance).
4. Authors have stated that “Compared to pure ZnO and pure CeO2, the composite materials S2 and S3 have a different Eg value, due to the confinement effects of the heterojunctions”. This needs to be discussed in more details. How the confinement effect is justified in your case?
5. Please explain why the degradation level decreases in higher contents of CeO2 (sample 4 compared to samples 3).
6. A revision in English is recommended, though overall English is fine.
Author Response
Response to comments of Reviewer 2 and list of changes made in the manuscript:
"Photocatalytic removal of antibiotics from wastewater using the CeO2/ZnO heterocatalyzers"
"Photocatalytic removal of antibiotics from wastewater using the CeO2/ZnO heterostructures"
We thank the reviewer for comments and suggestions [recommendations, proposals] to improve the academic quality of our manuscript. We have carefully studied all reviewer comments and revised the manuscript to take them into account. The proposed modifications have been evidently marked in green in the revised manuscript. Details of our responses to reviewer comments are provided below. We hope you will find these revisions rise to your expectations.
The manuscript "Photocatalytic removal of antibiotics from wastewater using the CeO2/ZnO heterocatalyzers" represents some interesting aspects of highly functional hybrid photocatalyst heterostructures. In this work, CeO2/ZnO-based photocatalytic materials were synthesized by the sol-gel method in order to establish heterojunctions that increase the degradation efficiency of some types of antibiotics by preventing the recombination of electron-hole pairs. The synthesized materials were analyzed by XRD, SEM, EDAX FTIR and UV-VIS. Modification of bandgap structure is investigated and correlated to the catalysts properties. The authors have very perfectly represented the work with interesting data and discussions. The analyses are very interesting for the readers in this field of stuyd. I strongly recommend this manuscript for publication. The manuscript, however, requires some mandatory revisions before being considered for publication.
1.The introduction on similar works and more specifically on the implications of CeO2 and the implications of heterostructures for the catalytic activities needs a revision. I strongly recommend authors consult following manuscripts in their revision:
-Total Oxidation of Light Alkane over Phosphate-Modified Pt/CeO2 Catalysts. Environmental Science & Technology, 56 (13), 9661-9671, 2022,
doi: 10.1021/acs.est.2c00135
-Non-free Fe dominated PMS activation for enhancing electro-Fenton efficiency in neutral wastewater. Journal of Electroanalytical Chemistry, 928, 117062, 2023
doi: https://doi.org/10.1016/j.jelechem.2022.117062
Thank you for the suggestion, we have followed the instructions and filled in the introduction (line 39) and references 23, 24 (line 341-345).
2.I wonder if you have any data on the photocatalytic activity of your samples under visible light radiation.
The two antibiotics are stable to visible radiation, and even together with the synthesized materials, they do not degrade in the presence of visible radiation. In the UV-Vis spectra of the freshly prepared antibiotic solutions and of the solution kept (forgotten) in the laboratory for 3 months, no changes appeared.
3.In Figure 2, please report EDS data in one digit accuracy (5.2 for instance).
We thank you for your observation, we have corrected in figure 2.
4.Authors have stated that “Compared to pure ZnO and pure CeO2, the composite materials S2 and S3 have a different Eg value, due to the confinement effects of the heterojunctions”. This needs to be discussed in more details. How the confinement effect is justified in your case?
Thank you for your observation, we have completed it in the text. The electrons in the 4f orbitals of Ce interfere with the 3d electrons of Zn and the 2p electrons of O, resulting in the formation of a new band between VB and CB that changes the characteristics of the oxide mixture (lines 81-83, 179-182 and 219-221) (references 53,54).
5.Please explain why the degradation level decreases in higher contents of CeO2 (sample 4 compared to samples 3).
In the case of the pure CeO2 sample, the degradation efficiency decreases due to the high recombination speed of the electron-hole pairs. In samples S2 and S3 an intermediate energy band is formed, suggested in Figure 8, reconstructed (line 217), the appearance of this band at the separation boundary between the two oxides leading to improved optical properties (e.g. lower Eg value) (line 229-231).
6.A revision in English is recommended, though overall English is fine.
We made small corrections

Round 2
Reviewer 1 Report
Unfortunately, I am forced to note that the authors did not provide in the introduction what is the novelty of their work, because there are so many publications on the hydrothermal synthesis of ZnO with CeO2 heterojunctions and their photocatalytic activity. In addition, the work requires careful editing. For example, it is not clear why in Fig. 8 the same scheme is given twice. The text of the article should also be edited, even in the title there is a mistake "...heterojonction". In this form, the article cannot be accepted for publication. Revision of the article in the introduction section is required.
Author Response
Response to comments of Reviewer 1 and list of changes made in the manuscript:
Photocatalytic removal of antibiotics from wastewater using the CeO2/ZnO heterojunction
We would like to thank the reviewer for comments and suggestions for improving the scientific quality of our manuscript. We have carefully considered all the reviewer comments and have revised the manuscript in light of them. The suggested modifications were clearly marked in red in the revised manuscript. Details of our responses to reviewer comments are shown below. We hope you will find these revisions rise to your expectations.
Reviewer 1
Unfortunately, I am forced to note that the authors did not provide in the introduction what is the novelty of their work, because there are so many publications on the hydrothermal synthesis of ZnO with CeO2 heterojunctions and their photocatalytic activity. In addition, the work requires careful editing. For example, it is not clear why in Fig. 8 the same scheme is given twice. The text of the article should also be edited, even in the title there is a mistake "...heterojonction". In this form, the article cannot be accepted for publication. Revision of the article in the introduction section is required.
We thank the Reviewer 1 for these useful remarks. These corrections were done in the revised manuscript.
We revised the Introduction part in accord with reviewer recommendation.
All corrections were highlighted.
At the suggestion of Reviewer 1, in Introduction we provide novelty:
As we know, there are several available studies on CeO2/ZnO applications, as heterostructures as well as individually [20, 24, 32, 38]. But here, PVA was used as a dispersant for heterostructure management and the obtained material showed good re-sults for the photocatalytic degradation of two antibiotics, separately and in admixture, for low doses of UV radiation and for small amount of catalyst. The results of this study can serve as a starting point for further research on CeO2/ZnO materials.
In Fig. 8 the same scheme is given twice
Thank you for the suggestion, the Fig. 8 was revised.
